



# Estimation of Nocturnal Boundary Layer Height in the Central Amazon, Supported by Gas Concentration Profiles

Carla M.A Souza[1,2], Anne C.S. Mendonça[2], Hella Van Asperen[1], Flávio A. D'Oliveira[3], Santiago Botía [1], Luís G. N. Martins[4], Denisi H. Hall[5], Raoni A. Santana[6], Gilberto Fisch[7], Leonardo R. Oliveira[5], Jailson R. Mata[5], Ranyelli Figueiredo[2,5], Bruno T.T Portela[5], Carlos A. Quesada[5], and Cléo Q. Dias-Júnior[2,5]

[1] Max Planck Institute for Biogeochemistry, Jena, Thuringia, Germany
[2] National Institute of Amazonian Research (INPA), Graduate Program in Climate and Environment (CLIAMB), Manaus, Amazonas, Brazil
[3] Federal University of the State of Pará , Graduate Program in Environmental Sciences (PPGCA), Belém, Pará, Brazil
[4] Federal University of Santa Maria, Santa Maria, Rio Grande do Sul, Brazil
[5] National Institute of Amazonian Research (INPA), Large-Scale Biosphere-Atmosphere Program in the Amazon (LBA), Manaus, Amazonas, Brazil
[6] Federal University of Western Pará (UFOPA), Santarém, Pará, Brazil
[7] University of Taubaté, Taubaté, São Paulo, Brazil

**Correspondence:** Carla M.A Souza (calves@bgc-jena.mpg.de) and Cléo Q. Dias-Júnior (cleo.quaresma@inpa.gov.br)

**Abstract.**

The height of the nocturnal boundary layer ($h_n$) is a fundamental parameter for weather and climate prediction. However, because turbulent processes weaken at night, estimating $h_n$ remains challenging. In addition, our understanding of its variability is limited, especially due to the predominant use of indirect methods that do not always accurately reflect the physical definition of the boundary layer. In this study we used micrometeorological measurements collected at the Amazon Tall Tower Observatory, in central Amazon. These measurements enable the study of turbulent sensible heat flux ($H$) profiles from the canopy top up to 300 m above ground, from which $h_n$ can be defined. Our analysis focused on the seasonal differences between dry and wet periods for a La Niña year and an El Niño year. also, we explore how variations in $h_n$ affect the vertical distribution of CO and $CH_4$ concentrations. The results revealed significant variations, such as: largest values of $h_n$ were observed during the wet season of a year marked by the La Niña phenomenon ($\sim$270 m +/- 40 m), while smallest values of $h_n$ occurred in the dry season associated with El Niño ($\sim$100 m +/- 27 m). It was also observed that $h_n$ can act as a "barrier" to the entry or exit of air masses with high concentrations of CO and CH4. This study provides important insights into the variability of $h_n$ above the Amazon forest, with implications for improving parameterizations in atmospheric models.

**Keywords.** Amazon Forest, Nocturnal Boundary Layer, Turbulent Flux Profiles, Trace Gases

## 1 Introduction

The Atmospheric Boundary Layer (ABL) is the lower part of the atmosphere that is in direct contact with the thermal and mechanical forcings that act near the surface (Garratt, 1980; Keirsbulck et al., 2002). The height of the ABL ($z_i$) plays an





important role in the exchange of heat, moisture, gases and particles between the surface and the free atmosphere (Driedonks, 1982; Levi et al., 2020). Several studies have been carried out with the objective of estimating the height of the ABL, including
for remote regions such as the Amazon (Fisch et al., 2004; Guo et al., 2016; Carneiro and Fisch, 2020; Krishnamurthy et al., 2020; Dias-Júnior et al., 2022; Molero et al., 2022; Souza et al., 2023). However, the majority of the studies carried out so far have given special attention to the variability of $z_i$ during daytime, thus up to now there is little information about the variability of $z_i$ during nighttime (Carneiro and Fisch, 2020; Mendonça et al., 2025a).

The daytime variability of $z_i$ is well known. For example, in the southwestern Amazon, Fisch et al. (2004) showed that
$z_i$ values vary between pastures and forests, and observed that the boundary layer is deeper over pasture areas than above the forest. Carneiro and Fisch (2020) used 2 years of data collected during the Green Ocean Amazon (GoAmazon 2014/5) project campaign, carried out in the central Amazon, and observed that the maximum height of the convective boundary layer (CBL) was approximately 1,200 m in the wet season and approximately 1,600 m in the dry season. In addition to diurnal and seasonal variability, large-scale events such as El Niño and La Niña also influence the evolution of the boundary layer in the
Amazon (Carneiro and Fisch, 2020; Dias-Júnior et al., 2022; Souza et al., 2023). During El Niño, the anomalous warming of the equatorial Pacific intensifies the occurrence of drier conditions, favoring a deeper CBL (Dias-Júnior et al., 2022; Souza et al., 2023). In contrast, during La Niña, greater moisture convergence favors increased precipitation that tends to limit its growth (Souza et al., 2023). Dias-Júnior et al. (2022) used ceilometer data, also collected during the GoAmazon project, and showed that daytime $z_i$ values are maximum during the dry season in El Niño years.

$Z_i$ plays an important role in the dispersion of pollutants near the surface. Based on ten years of measurements at 1,500 stations in China, Su et al. (2018) observed that $z_i$ and particulate matter concentrations exhibit a negative correlation, indicating that deeper layers favor the dispersion of pollutants. In a complementary way, Miao et al. (2015) showed that the interaction between $z_i$ and local circulations strongly modulates air quality in the Beijing–Tianjin–Hebei region. They showed that in autumn and winter, when $z_i$ is generally shallower, and often combined with mountain breezes, it favors the accumulation of
pollutants in the region. Meanwhile, in spring and summer, with deeper $z_i$ and combined with the sea breeze, they promote more efficient dispersion conditions throughout the day.

On the other hand, the relationship between $z_i$ and pollutants has been shown to be bidirectional; Wang et al. (2019) observed that severe air pollution episodes occurred under conditions of weak winds, high humidity, and strong temperature inversions. The accumulation of aerosols reduced the incoming solar radiation and the sensible heat flux, which led to a weakening of
vertical mixing and a 44–56 % reduction in $z_i$ compared to clean days. Kulmala et al. (2023) in a study focused on boreal and Arctic regions within the framework of the Pan-Eurasian Experiment (PEEX), highlighted that the $z_i$ also controls the vertical mixing and accumulation of trace gases such as $CO_2$, $CH_4$, and reactive volatile compounds. They emphasized that variations in $z_i$ modulate near-surface concentrations of these gases, influencing local chemical reactivity and the radiative balance. Conversely, trace gases and aerosols can feedback on boundary-layer development by altering radiative fluxes, air
temperature, and stability conditions.

To date, few studies have been conducted that study the nocturnal boundary height ($h_n$) above the Amazon rainforest. Santos et al. (2007) using radiosonde data measured in the southwestern Amazon, showed that $h_n$ values were around 250 m. Dias-



Júnior et al. (2022) using data from different instruments, showed that $h_n$ values in the Central Amazon were 200 m. Recently, Mendonça et al. (2025a) used data from the ATTO site and showed that vertical profiles of $H$ offer an opportunity to obtain direct measurements of $h_n$, unlike other methodologies that provide indirect measurements, such as radiosondes, ceilometers, and temperature profiles, etc. They observed that $h_n$ varies according to atmospheric stability and local topography, with mean $h_n$ values around $\sim 80$ m under very stable conditions and $\sim 170$ m under near-neutral conditions.

Although vertical profiles of turbulent fluxes up to heights close to 325 m are uncommon at experimental sites worldwide, the ATTO project has been measuring such profiles (at 19 levels) since mid-2021. With these flux profiles it is possible to directly estimate $h_n$ throughout the nighttime period, including the early morning, since the height of the boundary layer remains below the top of the tower. Although the study by Mendonça et al. (2025a) has contributed to recent advances in estimates of $h_n$ values, the seasonal and interannual variability of $h_n$ in the Central Amazon remains poorly understood. Moreover, no studies so far have directly addressed how $h_n$ variability influences the near-surface concentrations of trace gases in the region. In view of these gaps, the main objective of this study is to investigate the seasonality and interannual variability of $h_n$ in the ATTO region. In addition, it seeks to explore how variations in $h_n$ affect the dispersion of selected trace gases in the region. To achieve this, we estimate $h_n$ using vertical profiles of the turbulent sensible heat flux. We believe that this work provides an important contribution to advancing knowledge in fields such as micrometeorology, aerosol dynamics, and greenhouse gas studies, all of which require precise descriptions of the structure of the Nocturnal Boundary Layer (NBL) (Santana et al., 2018; Botía et al., 2020; Barbosa et al., 2022; Franco et al., 2024).

## 2 Data and Methods

### 2.1 Experimental Site and Data Acquisition

The data used in this work were measured in 2022 and 2023 at the ATTO experimental site, located in the central Amazon, about 150 km northeast of Manaus, in the Uatumã Sustainable Development Reserve (SDR) (Figure 1a). The site is characterized by dense tropical forest with a relatively uniform canopy, whose tree crowns generally reach a height of around 37 m (Andreae et al., 2015; Gomes Alves et al., 2023). A detailed description of the site is provided by Andreae et al. (2015).

We used fast-response data (10 Hz) collected by sonic anemometers installed on two towers (located at $2.148°$ S, $59.0068°$ W, and $2.1448°$ S, $59.0008°$ W) that are 670 m apart (Dias-Júnior et al., 2019). The 81 m tower (INSTANT Tower) and the 325 m tower (ATTO Tower) are on a plateau, located at approximately 130 m above sea level (Chamecki et al., 2020). The sensors used included CSAT-3B models from Campbell Scientific, Inc. (Logan, USA) installed at 50 and 81 m on the INSTANT Tower, and 3D Ultrasonic Anemometer model 4.3830 series from Thies Clima3D (Göttingen, Germany) installed at 100, 127, 151, 172, 223, 247, 274, and 298 m on the ATTO tower, all heights positioned above the ground. The variables measured were air temperature and the three components of wind velocity ($u$, $v$, and $w$). Measurements from both towers were integrated into a single vertical profile, since previous studies did not identify significant differences between them (Mendonça et al., 2025a).

Relative humidity and air temperature were obtained by thermohygrometers model IAKM I-Series from Galltec-Mela (Bondorf, Germany), and atmospheric pressure was recorded by barometers model 61302V (Young, USA), both operating at 0.01 Hz



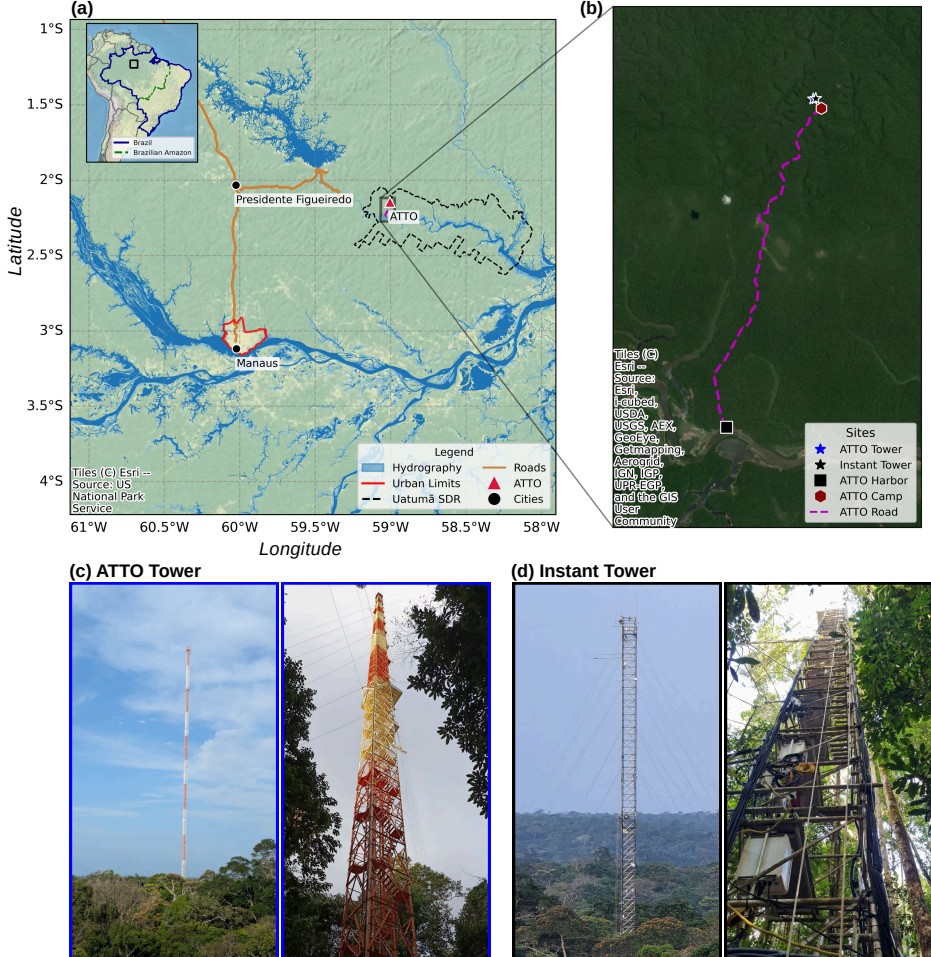

**Figure 1.** Geographic location and infrastructure of the Amazon Tall Tower Observatory (ATTO). a) Regional map showing the position of ATTO northeast of Manaus, within the Uatumã Sustainable Development Reserve (SDR); b) High-resolution zoom around the ATTO site based on Esri World Imagery, showing the ATTO Tower (blue star), Instant Tower (black star), ATTO harbor (black square), ATTO Camp (brown hexagon), and the ATTO access road (magenta dashed line); c) Photographs of the ATTO Tower (325 m) from ground perspective and structural details of its metallic frame; d) Photographs of the Instant Tower (80 m). All maps were generated using Python 3.11.6, in the WGS84 geographic coordinate system (EPSG:4326). Basemaps: Esri World Physical (panel a) and Esri World Imagery (panel b). Sources: Esri, TomTom, FAO, USGS | Powered by Esri.

at the same heights as the anemometers. Net radiation ($R_n$) was calculated as the sum of the shortwave radiation components (incoming and reflected), measured by a CMP21 pyranometer, and the longwave radiation (incoming and emitted), recorded by a CGR4 pyrgeometer (both Kipp & Zonen, Delft, Netherlands). These measurements were carried out at 81 m on the IN-STANT tower, at 0.01 Hz. CO and $CH_4$ concentrations were measured once per hour by a gas analyzer (FTIR Spectrometer)



installed at the foot of ATTO tower, sampling air at five heights: 42, 81, 150, 273, and 321 m van Asperen et al. (2023). All
       analyses considered the nighttime period from 00 to 09 UTC (20 to 05 local time; UTC−4), avoiding transition intervals.

## 2.2    Quality Control and Data Processing

As in other micrometeorological studies (Foken and Mauder, 2008; Starkenburg et al., 2016; Dias et al., 2023), the high-
frequency data were initially segmented into 30-minute intervals (totaling 18,000 measurements) for the application of quality
control (QC) procedures and tilt correction. QC was performed for each wind component and for the temperature measured
       by the sonic anemometers, following the protocols described by Vickers and Mahrt (1997); Zahn et al. (2016) and Dias et al.
       (2023). The QC process included verification of record integrity, detection of error indicators, identification and removal of
       spikes, as well as the application of stationarity tests. For a detailed description of the quality control protocol adopted at
       the ATTO Site, it is recommended to consult Zahn et al. (2016) and Dias et al. (2023). After QC, turbulence statistics were
calculated in 5-minute time windows.

$H$ values were obtained by the eddy covariance method (Webb et al., 1980), according to the equation: $H = \rho c_p \overline{w'T'}$, where
$\rho$ is air density (we assumed $1.225 \, \text{kg m}^{-3}$), $c_p$ is the specific heat of air at constant pressure ($\text{J kg}^{-1} \, \text{K}^{-1}$). The variables $T'$
and $w'$ represent the fluctuations of temperature and of the vertical wind component, respectively. The mean wind speed ($U_m$)
was calculated as $U_m = \sqrt{u^2 + v^2}$, where $u$ and $v$ are the zonal and meridional wind components, respectively. The potential
temperature ($\theta$) was calculated as $\theta = T_h (p_0/P)^{R_d/c_p}$, where $T_h$ is the air temperature measured by the thermohygrometer
       (Kelvin), $P$ is atmospheric pressure (hPa), $p_0$ is the reference pressure (1000 hPa), and $R_d$ is the gas constant ($287 \, \text{J kg}^{-1} \, \text{K}^{-1}$).
       Profiles of $H$ were used to estimate $h_n$, while $U_m$ and the vertical gradient of $\theta$ were used to discuss its variability.

Atmospheric stability was evaluated using the Monin–Obukhov stability parameter ($z/L$), where $L$ represents the Obukhov
length, determined as $L = -u_*^3 T/(kg\overline{w'T'})$, with $u_*$ the friction velocity, $k$ the von Kármán constant (0.41), $g$ the acceleration
due to gravity, and $\overline{w'T'}$ the kinematic sensible heat flux. The parameter $z/L$ was calculated considering the variables $u_*$,
       $\overline{w'T'}$, and $T$ at 50 m, similar to that performed by Cava et al. (2022).

## 2.3    Data Filtering and Classification

After QC and calculation of turbulence statistics, two filters were applied to the dataset of the 5 min. First, to avoid the influence
of extreme values, outliers in the $H$ measurements were identified by means of histograms by height layer. Based on the
observed distribution, the filtering limits were defined empirically, resulting in the following specific ranges: from 50 to 81 m
       ($-50$ to $0 \, \text{W m}^{-2}$), from 100 to 151 m ($-30$ to $10 \, \text{W m}^{-2}$) and from 172 to 298 m ($-20$ to $10 \, \text{W m}^{-2}$). Values outside these
       ranges were considered outliers and removed from the analysis. The application of differentiated limits reflects the expected
       variation of $H$ among the layers, being more intense in the lower layers due to direct interaction with the canopy, and smoother
       at higher altitudes (Gao et al., 1989; Shaw and Schumann, 1992; Mendonça et al., 2025a).
The second filter considered very stable atmospheric conditions, characterized by very low sensible heat fluxes above the
       canopy, indicative of minimal energy exchange, configuring a strongly decoupled regime (Cava et al., 2022). In these situations,
       the vertical profile of $H$ does not present a typical structure (higher values near the canopy and a decrease with height), but





**Table 1.** Total number of 5-minute profiles and percentage of profiles excluded after data filtering.

| Year | Period | Total number of profiles | Excluded profiles (%) |
|------|--------|--------------------------|------------------------|
| 2022 | Wet    | 6425                     | 13                     |
| 2022 | Dry    | 7811                     | 27                     |
| 2023 | Wet    | 7080                     | 31                     |
| 2023 | Dry    | 6603                     | 28                     |

rather an irregular pattern, which makes the reliable determination of $h_n$ difficult. So, cases in which the $H$ values at $50\,\mathrm{m}$ were between 0 and $-6\,\mathrm{W\,m^{-2}}$ were excluded, a range identified as the most frequent based on the data distribution under
very stable conditions ($1 \leq z/L \leq 10$, such as Mendonça et al. (2025a)).

Subsequently, the data were classified by seasonal period (Dry and Wet) in each year, with 2022 associated with La Niña and 2023 with El Niño (Table 1). According to the NOAA Climate Prediction Center, La Niña conditions persisted from late 2021 to early 2023, whereas a strong El Niño developed around May 2023 and lasted until early 2024. In the central Amazon, La Niña typically brings wetter conditions and slightly cooler temperatures, while El Niño tends to produce drier and warmer
conditions, particularly during the Dry season, due to reduced convective activity and changes in large-scale atmospheric circulation (Marengo et al., 2018; Jimenez et al., 2018).

The Dry period was defined as the months of August, September, and October, and the Wet period encompassed the months of January to April. The inclusion of January is justified by the scarcity of data available in April, which could compromise the robustness of the analyses. Considering that January still falls within the region's Wet season, its inclusion increases the
representativeness of the data. Moreover, Dias-Júnior et al. (2025) showed that January remains among the rainiest months for ATTO's area. This division aligns with regional seasonal precipitation patterns in the Central Amazon described by Marengo et al. (2001) and, for ATTO specifically, by Dias-Júnior et al. (2025). From this classification, hourly means of all analyzed variables were calculated at their respective measurement heights.

## 2.4 Estimation of the Height of the Nocturnal Boundary Layer

With the hourly mean profiles of $H$ obtained after filtering and classification, it was possible to estimate $h_n$ for each hour of the nighttime period, separately for the Dry and Wet seasons of 2022 and 2023. Since the sensible heat flux is greater near the canopy and decreases with height, $h_n$ was identified as the height at which $H$ was less than $5\,\%$ of the $H$ value measured near the canopy ($50\,\mathrm{m}$) (Lenschow et al., 1988; Mendonça et al., 2025a). This turbulence-based approach is consistent with the fundamental definition of the atmospheric boundary layer. To allow comparison among profiles, the $H$ values at all heights
were normalized by the value measured at $50\,\mathrm{m}$.

In addition to the seasonal and hourly analyses, we selected two nights as case studies to investigate the relationship between variability in $h_n$ and the concentrations of CO and $CH_4$ above the forest canopy: one with a shallow $h_n$ layer ($h_n \sim 80\,\mathrm{m}$) and





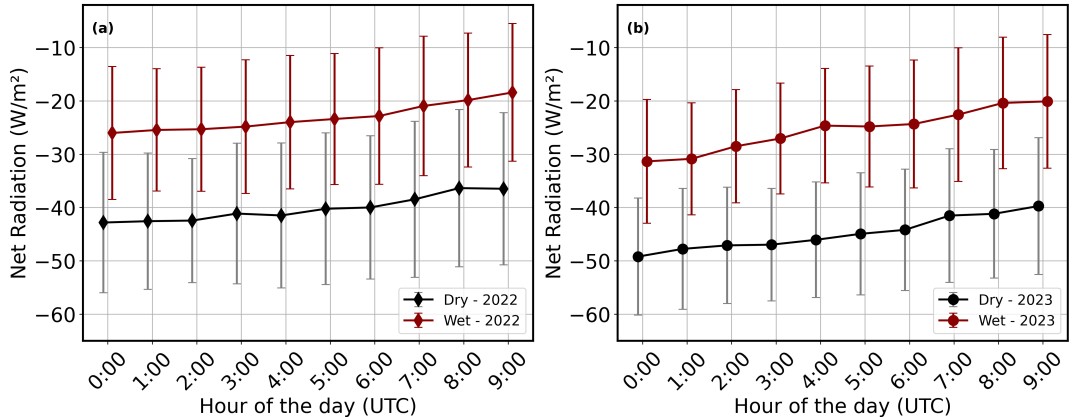

**Figure 2.** Net radiation (Rn) for the Wet season (in red) and the Dry season (in black) during a) 2022 and b) 2023. The bars indicate the hourly standard deviation.

another with a deeper layer ($h_n \sim 150\,\text{m}$). The idea here was to investigate to what extent $h_n$ affects the temporal and spatial gas concentration variations along the tower.

## 3 Results and Discussion

### 3.1 Atmospheric Stability at the ATTO Site

Figure 2 shows the hourly mean values of $R_n$ for the Wet and Dry periods in the years 2022 and 2023. It is observed that both the Wet and Dry seasons of 2022 presented $R_n$ values with larger magnitude than those observed in the Wet and Dry seasons of 2023. In addition, it is noted that $R_n$ values for the Dry season are greater than those observed in the Wet season, in both years. It is known that cloud cover has a direct relationship with $R_n$ values (Barber and Thomas, 1998; Von Randow et al., 2004; De Oliveira et al., 2016). Greater cloud cover is associated with lower radiative loss, that is, smaller $R_n$ values. Therefore, in the Dry season, where cloud cover is lower, there is greater radiative loss. The same reasoning applies to the year 2023, an El Niño year, a period characterized by lower cloud cover over central Amazon (Jimenez et al., 2018; Restrepo-Coupe et al., 2023). Atmospheric stability also varies systematically between seasons. The Wet period is less stable (smaller $z/L$), while the Dry period shows stronger stability due to reduced cloud cover. Lower cloudiness limits wind generation and mechanical turbulence, reinforcing stable conditions. El Niño in 2023 enhanced this effect (Figure S1).

### 3.2 Seasonal Mean Cycle of NBL Height

Figure 3 presents the normalized vertical profiles of sensible heat ($H/H_{50}$). It is noted that the turbulent heat fluxes cease at different heights when comparing the Dry and Wet seasons, which indicates important seasonal variations of $h_n$. During the





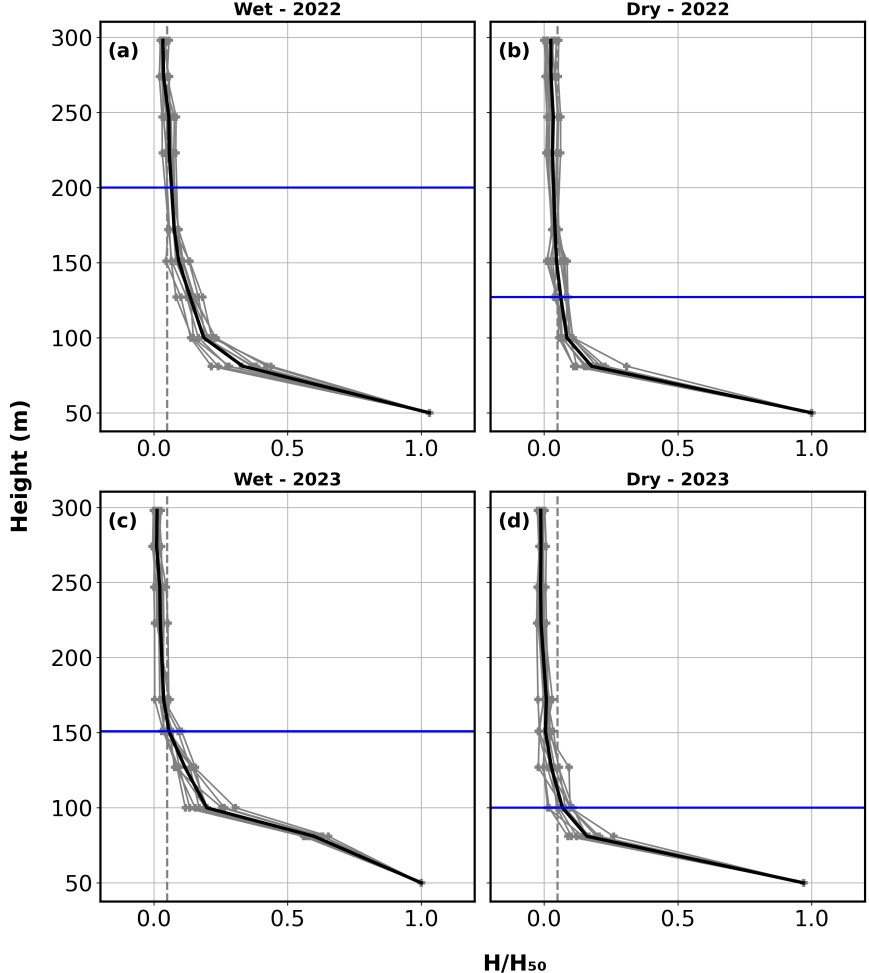

**Figure 3.** Hourly profiles of normalized sensible heat flux ($H/H_{50}$) for (a) Wet season 2022, (b) Dry season 2022, (c) Wet season 2023, and (d) Dry season 2023. Grey curves represent the individual hourly nighttime profiles, illustrating the variability in the vertical distribution of $H/H_{50}$ throughout the night. The bold solid curve in each panel shows the seasonal mean profile. The vertical dashed grey line marks the 5 % threshold ($H/H_{50} = 0.05$), and the horizontal blue line marks the seasonal mean height at which this threshold is reached, interpreted as the $h_n$.

Wet period, turbulence extends up to approximately $200 \pm 33$ m in 2022 and $150 \pm 24$ m in 2023, while in the Dry period this height is reduced to about $120 \pm 28$ m in 2022 and $100 \pm 21$ m in 2023.

    Figure 4 presents the hourly mean values of $h_n$ for the Wet and Dry periods for the two years analyzed here. It is noted that in 2022 the $h_n$ values show annual and seasonal variations, being 274 m at the beginning and 151 m at the end of the night during the Wet season, while in the Dry season the values range between 223 m (beginning of the night) and 100 m (end of the night). In 2023, the seasonal variation of $h_n$ is smaller between the seasons, ranging between 172 m at 00:00 UTC and 100 m




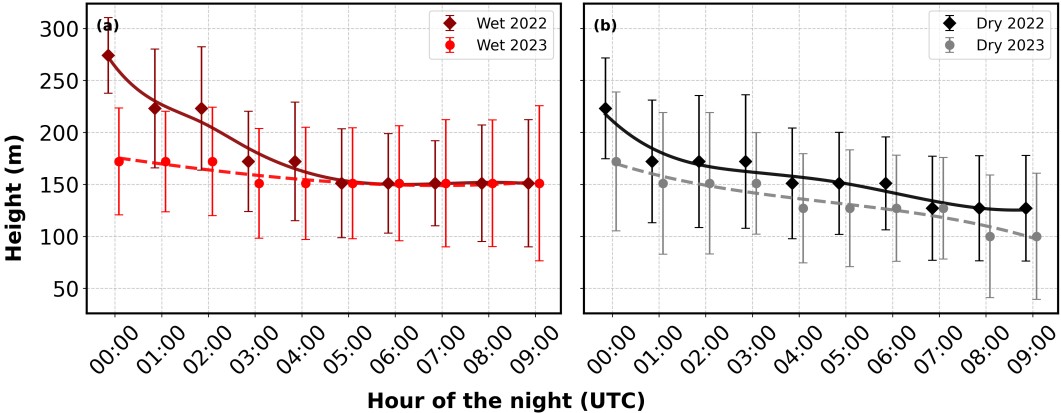

**Figure 4.** Mean values and standard deviation of $h_n$ for (a) the Wet period of 2022 and 2023 and (b) the Dry period of 2022 and 2023. The curves represent smooth fits obtained by spline between the mean $h_n$ values.

at 09:00 UTC. This behavior is associated with the variability of turbulence and stability (Figure 3 and Figure S1). In addition, it is observed that, regardless of the season or the year, the $h_n$ values are higher at the beginning of the night and decrease progressively. Also in Figure 4, it is possible to observe that the hourly variation of $h_n$ is more intense in the Wet season of 2022, given that the thermal gradient is weaker in this period, especially in the first hours of the night (Figure S2). In this way, mechanical turbulence keeps the NBL deeper. As the night advances, the thermal gradients intensify (Figure S2) and cause destruction of turbulence; with this, $h_n$ decreases in all periods, and more intensely in the Wet season of the La Niña year.

The differences observed in $h_n$ values between the Wet and Dry seasons and between the years 2022 and 2023 can be explained by the distinct radiative and thermodynamic conditions between the periods. During the Dry season, the atmosphere exhibits greater nocturnal radiative loss (Figure 2), which favors pronounced surface cooling and, consequently, the intensification of atmospheric stability (Figure S1). This condition helps inhibit turbulence and reduces the efficiency of the vertical transport of sensible heat. In the Wet period, greater cloud cover and high humidity attenuate radiative loss, resulting in a less stable and more turbulent atmosphere, which allows greater vertical mixing and more expressive $H$ fluxes, and this extends to higher levels. The same reasoning applies to the differences between the La Niña year (2022) and the El Niño year (2023). That is, greater cloud cover (like 2022) results in greater turbulent activity, especially in the Wet season of 2022.

Previous studies showed higher $h_n$ values, above 250 m; however, they used indirect methods to estimate $h_n$ (Carneiro et al., 2021; Dias-Júnior et al., 2022). More recently, Mendonça et al. (2025a) used the same dataset as in this study and showed that $h_n$ is lower. They showed that the mean $h_n$ values varied between 223 m and 81 m, and this variation depended on surface roughness (forest + topography). However, in the work of Mendonça et al. (2025a), attention was not given to the hourly variability of $h_n$, as is being shown here.



### 3.3 Temporal Variability of NBL Height and Its Relationship with Greenhouse Gas Concentrations Near the Surface: Case Studies

In general, $h_n$ shows a gradual decrease throughout the night (Fig. 4) However, this pattern can vary depending on turbulence intensity, thermal stratification, and the atmospheric stability conditions (Figures 2 and S1). These changes in $h_n$ height can be reflected in the vertical profiles of trace gases which are measured along the tower. In order to explore this relationship we refer to Figure 5, which shows two case studies: one in which the $h_n$ values remained low throughout the entire night (18 August 2022), and another in which the $h_n$ values showed a progressive increase throughout the night (25 August 2022). The two nights selected exhibit quite different thermodynamic characteristics. The wind profiles (Figure 6) show the occurrence of low-level jets (LLJs) on both nights, but with distinct characteristics. On the night of 18 August, the jet nose (maximum speed in the wind profile) was located at a height that oscillated around 80 to 100 m, while on the night of 25 August the jet nose was located around 170 m. Another important difference is observed in the dominant wind direction during the two nights. During the night of 18 August, the predominant wind direction was from the northeast at the lowest levels and from the southeast at the highest levels (Fig. 7 a). During the night of 25 August, the wind direction ranged between north and northeast from the top of the canopy up to 298 m. According to Mendonça et al. (2025b), the jet nose (maximum speed in the wind profile) can serve as an indicator of $h_n$, which is consistent with what we observe in our case studies.

Figure 8 shows the CO and CH$_4$ concentrations for the two case studies. During the night of 18 August, the propagation of an air plume rich in CO and CH$_4$ that reached the tower was observed. However, this air mass was only detected by the inlets located at the highest heights (321 and 273 m), producing strong vertical differences in the tower CO and CH$_4$ profiles. The observation of elevated CO and CH$_4$ concentrations at higher altitudes, together with the different flow direction above the $h_n$ (Figure 5 and 7), suggests that these air masses originated from more distant sources. Air arriving from the SE–S sector (above the NBL, Figure 5) likely transported high concentrations of CO and CH$_4$, possibly associated with regional biomass burning plumes (Andreae et al., 2015; Van Asperen et al., 2024) (Figure S3). In addition, it is believed that the plume did not penetrate down to canopy level (50 and 81 m) because the shallow $h_n$ that night acted as a barrier (Figure 5). Meanwhile, air masses originating from the N–NE (below the NBL, Figure 5 and 7) exhibited lower CO and CH$_4$ concentration. We hypothesize that the height of 150 m is in the transition between the two layers, which explains why the CO and CH$_4$ concentrations coincide partly with those observed at 42 and 81 m. It is also noted that around 07:00 UTC the CO and CH$_4$ concentrations at the higher heights begin to decrease, reaching values similar to those observed at the lower heights at 09:00 UTC, which could indicate that the tower is leaving the concentration plume, caused by a change in wind direction at higher heights which shifted from SE–S to N–NE. To clarify, the layers remain decoupled, but the flow in both layers comes from the same direction and is no longer influenced by air masses with elevated CO and CH$_4$ concentration (Figure S3a).

On the night of 25/08, CO concentrations were higher than those observed on 18 August, indicating a multi-day pattern in which CO levels were generally elevated across the region (Figure S4), as more often observed in the region (Van Asperen et al., 2024). At 00:00 UTC on 25/08 the highest CO and CH$_4$ concentrations were observed at the heights closest to the forest canopy; thereby different than at 18/08 (Figure 8 b). Although plumes typically arrive first above the canopy, this apparent





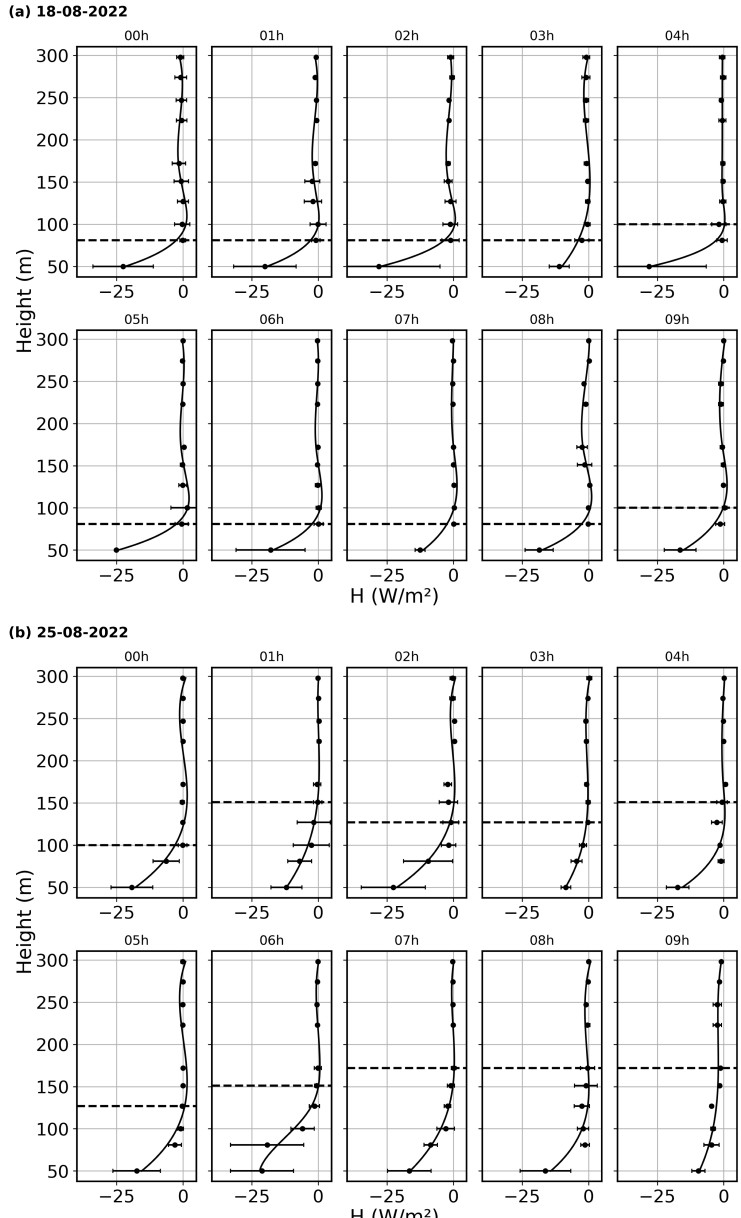

**Figure 5.** Hourly mean profiles of sensible heat flux (black curve), standard deviation, and the corresponding $h_n$ (horizontal dotted lines) for a) 18/08/2022 and b) 25/08/2022.

inversion can be explained by the slower response of air within the canopy to atmospheric changes. Elevated air from previous
days (before 25 August; see Figure S4) may have penetrated the canopy, and while the passing plume quickly removed high
concentrations from the upper layers, ventilation near the canopy occurred more slowly, resulting in higher concentrations





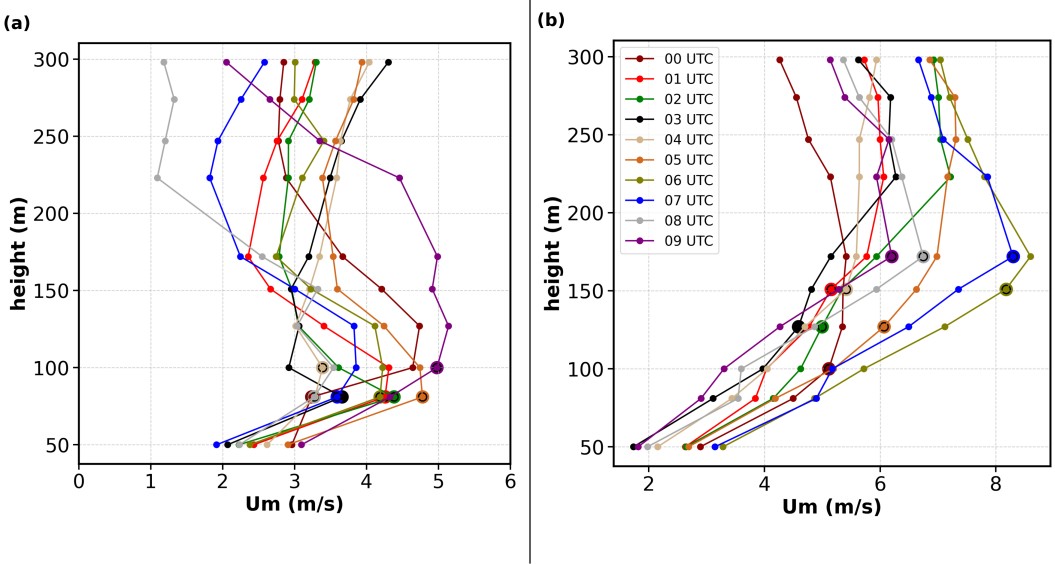

**Figure 6.** Hourly mean wind speed profiles for the nights of a) 18/08/2022 and b) 25/08/2022. The different colors indicate the different hours. The larger hollow circles mark the hourly estimates of the $h_n$, determined from the sensible heat flux profiles.

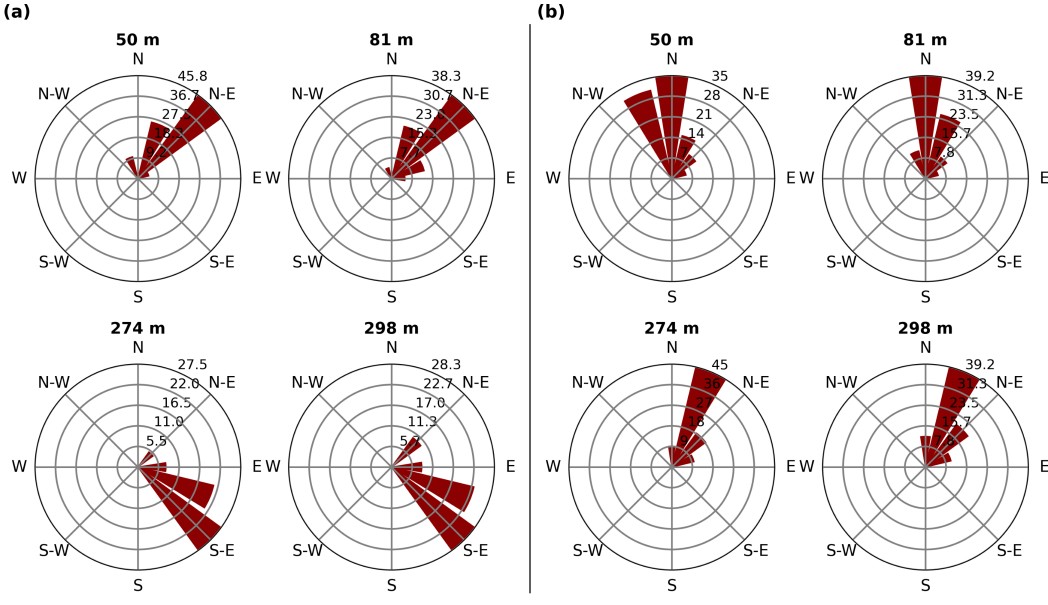

**Figure 7.** Wind direction at four distinct heights for the nights of a) 18/08/2022 and b) 25/08/2022.

at lower levels. Therefore, at 00:00 UTC, it appears that there is more CO and $CH_4$ at the lower levels, but this may simply reflect a delayed residual effect of a previous plume that has not yet been flushed out from the lower layers of the NBL. This is





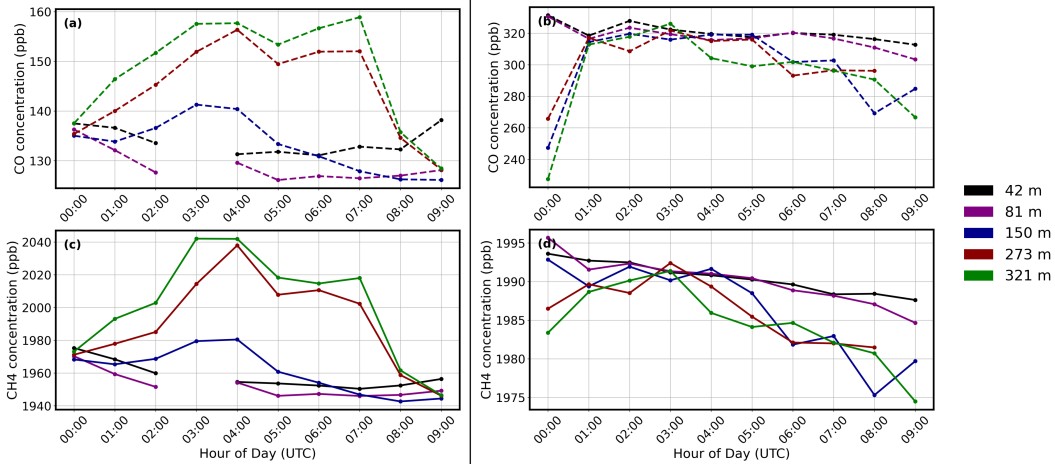

**Figure 8.** Vertical profiles of CO and $CH_4$ for the nights of 18 August 2022 and 25 August 2022. CO is shown in panels (a) and (b), and $CH_4$ in panels (c) and (d). Dashed lines indicate CO and solid lines indicate $CH_4$.

consistent with the difference in wind speed between heights, about $3\,\mathrm{m\,s^{-1}}$ at $50\,\mathrm{m}$ and $5\,\mathrm{m\,s^{-1}}$ at $150\,\mathrm{m}$ (Figure 6), suggesting
slower air renewal at the lower levels.

One hour later (at 01:00 UTC) there is practically no difference between the CO and $CH_4$ concentrations at all measurement levels, remaining so until around 03:00 UTC. This behavior suggests that a new plume arrived at 01:00 UTC. It should be noted that at 03:00 UTC the CO and $CH_4$ concentrations at $321\,\mathrm{m}$ start to decrease, while at the lower levels there is a lag in this decrease. From 05:00 UTC the CO and $CH_4$ concentrations begin to decrease rapidly at the higher heights, while at the lower
levels they decrease much slower. Overall, these two case studies highlight that the $h_n$ is reflected in the vertical CO and $CH_4$ concentration profiles, thereby supporting our estimated $h_n$. When there is strong decoupling between the NBL and the layer above, air within the NBL can become trapped: the CO and $CH_4$ plume may be unable to enter the NBL, as on 18 August, or may be slow to ventilate out, as observed on 25 August.

## 4    Conclusions

This study presented the hourly variability of $h_n$ heights during the Wet and Dry periods, based on turbulent data measured at the ATTO site (central Amazon) in a La Niña (2022) and an El Niño (2023) year. The results showed that $h_n$ is controlled by differences in net radiation and atmospheric stability between the seasons. Higher mean values were found in the Wet season under La Niña ($\sim 270 \pm 40\,\mathrm{m}$), while the lowest values occurred in the Dry season associated with El Niño ($\sim 100 \pm 27\,\mathrm{m}$). During the El Niño year, $h_n$ varied from approximately $223\,\mathrm{m}$ at the beginning of the night to about $100\,\mathrm{m}$ at the end of the
night. In contrast, in 2022, $h_n$ ranged from roughly $274\,\mathrm{m}$ at the start of the night to $151\,\mathrm{m}$ at the end. Seasonal variability was also more evident at the beginning of the night in 2022 compared to 2023.



Based on two case studies with contrasting conditions, the visibility of $h_n$ on nighttime CO and CH$_4$ concentrations above the canopy was explored: the first case study (18 August 2022) demonstrated a weak turbulence and a low $h_n$, and the second case study (25 August 2022) a stronger turbulence and a $h_n$ progressively growing throughout the night. The case studies further revealed that when winds come from a source region (southeast sector), as on 18 August, the CH$_4$ and CO peaks are primarily determined by wind direction. In this situation, $h_n$ defines the height limit where decoupling occurs, leading to strong vertical gradients (50 ppb for CH$_4$ and 10 ppb for CO). Conversely, when no distinct plume is present (25 August), the differences in concentration across heights are mainly controlled by wind speed, resulting in smaller gradients than in the first case.

*Author contributions.* Conceptualization: CMAS, ACSM, CQDJ, and HVA. Data curation: CQDJ, LGNM, HVA, and FAD. Formal analysis: CMAS, HVA, and CQDJ. Funding acquisition: CQDJ and BTTP. Methodology: CMAS and ACSM. Project administration: CQDJ, BTTP, and CAQ. Software: CMAS. Supervision: CQDJ and SB. Validation: CMAS and HVA. Writing (original draft preparation): CMAS and CQDJ.Writing (review and editing): CMAS, SB, HVA, GF, FAD, ACSM, RF, LRO, JRM, and DHH.

*Competing interests.* The authors declare that they have no competing interests.

*Acknowledgements.* This study is part of the Amazon Tall Tower Observatory (ATTO) project, which is supported by the German Federal Ministry of Education and Research (BMBF; contracts 01LB1001A and 01LK1602A), the Brazilian Ministry of Science, Technology and Innovation (MCTI/FINEP; contract 01.11.01248.00), and the Max Planck Society (MPG). Additional institutional support is provided by the Amazonas State Research Foundation (FAPEAM), the São Paulo State Research Foundation (FAPESP), the State University of Amazonas (UEA), the National Institute for Amazonian Research (INPA), the LBA Program, and the Secretariat of Sustainable Development (CEUC/RDS Uatumã). C. M. A. Souza acknowledges financial support from the Coordenação de Aperfeiçoamento de Pessoal de Nível Superior – Brasil (CAPES) through the Program in Climate and Environment (INPA/UEA; grant 88887.820693/2023-00). C. Q. Dias-Júnior acknowledges support from the Conselho Nacional de Desenvolvimento Científico e Tecnológico (CNPq; processes 440170/2022-8, 406884/2022-6, and 307530/2022-1). This study was also financed in part by CAPES – Finance Code 001 (grant 88881.933987/2024-01). We sincerely thank all partners involved in the long-term operation of the ATTO site for their continuous support. In particular, we acknowledge the essential logistical assistance provided by Roberta de Souza, Wallace Rabelo Costa, Nagib Alberto de Castro Souza, Amauri Rodrigues, Valmir Ferreira, and Antonio Huxley. We further thank the LBA micrometeorology group for their sustained commitment and technical efforts that made this work possible.



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
