# Peer review of "Estimation of Nocturnal Boundary Layer Height in the Central Amazon, Supported by Gas Concentration Profiles"

_EGUsphere, 2025_

## Referee Comment (RC1)

Review
*Estimation of nocturnal boundary layer height in the central Amazon,*
*supported by gas concentration profiles*

This manuscript addresses a timely and potentially interesting topic of the dynamics of the nocturnal boundary layer and the impact on the gas concentration profiles in a remote location of the Amazonia. The meteorological and atmospheric composition data is very comprehensive. However, the analysis and interpretation remain largely descriptive and do not sufficiently engage with the underlying physical mechanisms or relevant observational and theoretical studies. Substantial additional work would be required to strengthen the physical interpretation of the dynamics of the Amazonian Nocturnal Boundary Layer (ANBL) and fully exploit the observations. Given the extent of revisions needed, we recommend rejection at this stage, with the encouragement that the authors resubmit a thoroughly revised manuscript after deeper physical and observational analysis. In the following, we provide several points where the analysis could be further elaborated to establish clearer physical causality.

1. The study could benefit from an extended climatology of the stability of the ANBL. Based on only one La Nina year (2022) and one El Nino year (2023) it is hard to explain which part of the variability can be contributed to the ENSO and which part to the annual variability. With an elaborate description of the yearly variation, the ENSO effect could be clearly quantified.

2. In relation to this a clearer identification and quantification of the main drivers would substantially strengthen the manuscript. For example, Figure 2 presents longwave radiative cooling, but its role in enhancing nocturnal stratification is not sufficiently discussed. Further elaboration on how radiative cooling contributes to the stability of the nocturnal boundary layer would be valuable. Similarly, in lines 200–205, the manuscript discusses shear associated with changes in wind direction, which is a well-known mechanism for turbulence generation. Quantifying the magnitude of this shear using the available observations, and assessing its relative importance in sustaining turbulence and shaping the nocturnal boundary layer structure, would greatly improve the physical interpretation. We think here it can be convenient to connect and relate their findings with recent work of Mahrt and Acevedo (Boundary-layer Meteorology 187:141–161, 2023 and references therein)

3. In the results, Figure 4, which presents the nocturnal boundary layer height under wet and dry conditions for 2022 and 2023, would benefit from further elaboration. While the results are interesting, the discussion remains largely descriptive. A deeper analysis is needed to explain the origin of the observed differences. In particular, it would be valuable to discuss which physical processes may drive these contrasts and whether they can be linked to other controlling variables. In particular the role of locally driven processes such as stability, turbulence, surface fluxes, or large-scale meteorological conditions, that influence the ANBL.

4. In the same line, and perhaps to support their analysis, perhaps it is interesting to study if there are current formulations (representations) or other methods to estimate the NBL height that are able to reproduce the observational dynamic evolution.

5. In section 3.3 the role of the canopy deserves more explicit attention. Given the observational setup, the authors are in a unique position to assess how the forest canopy influences the

structure and dynamics of the nocturnal boundary layer (see for instance line 179 and 225). Although the results (line 223-228) conclude that the canopy plays a role, this influence is not sufficiently specified or supported by analysis. A more detailed discussion, ideally supported by quantitative evidence, of the mechanisms through which the canopy affects nocturnal stratification, turbulence, or mixing would substantially strengthen the manuscript.

6. Finally, additional background information would strengthen the results in Figure 8, which compared to two stability cases. More emphasis should be placed on the different background concentrations during the past days of the gases and their effect on the current nocturnal gas exchange. In addition, to truly understand the effect of stability on trace gas concentrations, it would be valuable to compare two nights with similar wind and background trace gas conditions. Especially, because previous papers highlight the complicated CH4 signal at ATTO with winds from the South-East (Botia et al., 2020).